# An Edible Oil Enriched with Lycopene from Pink Guava (*Psidium guajava* L.) Using Different Mechanical Treatments

**DOI:** 10.3390/molecules27031038

**Published:** 2022-02-03

**Authors:** Catalina Gómez Hoyos, Angélica Serpa Guerra, Shaydier Argel Pérez, Jorge Velásquez-Cock, Mariana Villegas, Piedad Gañán, Robin Zuluaga Gallego

**Affiliations:** 1Programa de Ingeniería en Nanotecnología, Universidad Pontificia Bolivariana, Medellin 050004, Colombia; catalina.gomezh@upb.edu.co (C.G.H.); shaydier.argel@upb.edu.co (S.A.P.); jorgeandres.velasquez@upb.edu.co (J.V.-C.); mariana.villegasb@upb.edu.co (M.V.); 2Facultad de Ingeniería Agroindustrial, Universidad Pontificia Bolivariana, Medellin 050004, Colombia; angelicamaria.serpa@upb.edu.co; 3Facultad de Ingeniería Química, Universidad Pontificia Bolivariana, Medellin 050004, Colombia; piedad.ganan@upb.edu.co

**Keywords:** lycopene, green process, mechanical treatments, sunflower oil

## Abstract

According to the regulations of the United States Food and Drug Administration (FDA), organic solvents should be limited in pharmaceutical and food products due to their inherent toxicity. For this reason, this short paper proposes different mechanical treatments to extract lycopene without organic solvents to produce an edible sunflower oil (SFO) enriched with lycopene from fresh pink guavas (*Psidium guajava* L.) (FPGs). The methodology involves the use of SFO and a combination of mechanical treatments: a waring blender (WB), WB+ high-shear mixing (HSM) and WB+ ultrafine friction grinding (UFFG). The solid:solvent (FPG:SFO) ratios used in all the techniques were 1:5, 1:10 and 1:20. The results from optical microscopy and UV–vis spectroscopy showed a correlation between the concentration of lycopene in SFO, vegetable tissue diameters and FPG:SFO ratio. The highest lycopene concentration, 18.215 ± 1.834 mg/g FPG, was achieved in WB + UFFG with an FPG:SFO ratio of 1:20. The yield of this treatment was 66% in comparison to the conventional extraction method. The maximal lycopene concentration achieved in this work was significantly higher than the values reported by other authors, using high-pressure homogenization for tomato peel and several solvents such as water, SFO, ethyl lactate and acetone.

## 1. Introduction

Lycopene is the most efficient singlet oxygen quencher among carotenoids, with health benefits associated with its consumption [1]. A growing body of clinical evidence supports its role as a nutrient with important health benefits since it appears to provide protection against a broad range of cancers [1,2,3]. This led to a copious amount of research focused on extracting lycopene for its use as a novel food ingredient [4]. Despite its health benefits, lycopene extraction involves the use of organic solvents such as hexane, acetone and ethyl acetate, among other nonpolar compounds [5,6,7,8]. According to the regulations of the United States Food and Drug Administration (FDA), this organic solvent should be limited in pharmaceutical and food products due to its inherent toxicity [9].

In accordance with the above, it is necessary to use other extraction methods. A possible alternative is edible oils, such as olive or sunflower oil (SFO) [10,11,12]. Then, the extraction of carotenoids with oils appears as an innovative opportunity to develop sustainable, efficient and cost-effective processes for the successive supply of natural lycopene. Authors such as Kunthakudee, Sunsandee, Chutvirasakul and Ramakul (2020) [13] used different oil types and reported the following in descending order of extraction efficiency: coconut oil > soybean oil > olive oil > palm oil > SFO, with approximately 80% of the maximum of lycopene in coconut oil [13]. It has been reported that carotenoids have a relatively high bioaccessibility in mixed micelles generated from long-chain triglycerides (LCTs) [14,15]. In addition, SFO is highly available on the market as the fourth most-produced vegetable oil and the third most-produced oilseed meal in the world among protein feed sources [16]. Therefore, SFO appears to be promising for the extraction of lycopene.

Antioxidants, in particular, lycopene, are found in the inner bodies of cells, i.e., vacuoles and lipid vesicles of plants such as tomatoes, watermelons and pink guavas [8]. Therefore, to release lycopene from the cell structure and transfer it to SFO, it is necessary to break the cell wall by means of an ultrasonic [12] or mechanical treatment [17,18,19], followed by a solid–liquid extraction. Mert et al. (2012) [19] used microfluidization (MF) to process ketchup mixes, with the aim of increasing the lycopene concentration. Results indicated that higher MF pressures resulted in higher detectable lycopene content, between 14.9 ± 1.2 and 28.2 ± 2.2 mg of lycopene/100 g wet wt., for the range of the pressures tested between 200 and 2000 bar [19].

The combination of edible oil and mechanical treatment to extract lycopene has been gaining attention since 2019 [13,17,18]. In a recent publication, Juric et al., 2021, proposed a novel methodology to extract lycopene from tomato by-products and produce lycopene emulsions [17]. In this study, the authors used high-shear mixing (HSM) as a pre-treatment to form an emulsion in the presence of water and SFO, followed by a sieving stage to separate seeds. The final step in the process was the high-pressure homogenization treatment (100 MPa, up to 10 passes) of tomato residues in the presence of water and SFO, which was able to promote the extraction of bioactive compounds concurrently with the formation of an O/W emulsion stabilized by the micronized residues. The authors reported the highest concentration 3.94 ± 0.12 mg of lycopene/g cream (oily phase) at 5 passes through a high-pressure homogenizer (HPH) at 100 MPa [17].

Other mechanical techniques used to extract biocompounds from vegetable tissue can be combined to extract lycopene, such as a waring blender (WB) [20], high-shear mixing (HSM) [18] and ultrafine friction grinding (UFFG) [21]. The last one is a nanotool with a top-down approach. It involves breaking down a bulk material into micro- and nanosized structures, which can contribute to the disruption of the plant cells, enhancing the release of intracellular compounds [21,22].

According to the opportunities provided by mechanical techniques and solid–liquid extraction using edible oil as a solvent for lycopene isolation, this research proposed a different mechanical treatment to produce an SFO enriched with lycopene from Colombian pink guava (*Psidium guajava* L.). For this purpose, three different mechanical treatments were evaluated. The effect of each mechanical treatment on lycopene extraction is associated with many operational variables, such as rpm, processing time and FPG:SFO ratio. This paper defines a set of operation variables based on the experience of authors for the use of this technique to extract other biomolecules [20,21,22,23,24] and is focused on evaluating the effect of FPG:SFO on lycopene extraction for three different mechanical processes. Homogeneous mixtures of 1:5, 1:10 and 1:20 SFO: FPG ratios were prepared using a WB followed by an HSM or UFFG process. The obtained SFO samples were analyzed by light microscopy and UV–vis spectroscopy, with the aim of identifying the mechanical treatment and operational conditions that reach higher lycopene extraction. 

## 2. Results and Discussion 

Lycopene is the predominant carotenoid in pink guava, and it is responsible for its red coloration [22] (Figure 1a). The lycopene content of FPG can vary considerably with the variety of pink guava, its ripening stage, extraction and quantification methods [25]. Therefore, as mentioned in Section 3, all the fruits used in this work are of the *Psidium guajava* L. variety at the same ripening stage. Lycopene crystals with a reddish coloration can be identified as they are enclosed tightly inside plant cells [25] (Figure 1b). Therefore, to extract a nonpolar compound such as lycopene from the cell structure and transfer it into a solvent, it is necessary to break the cell wall [8]. Ultrasound is a technique commonly used to break down the cell wall and is used in combination with hexane:ethanol:acetone (2:1:1) to extract lycopene [5,26,27,28]. Using this methodology, lycopene in FPG was 27.66 ± 0.62 mg/g FPG.

The combination of ultrasound and SFO was evaluated by Rahimi and Mikani [12]. They reached a lycopene extraction yield of around 87.25% contrasted to conventional solvent extraction (CSE) (104.85 mg lycopene per 100 g) [12]. However, ultrasound is a hardly scalable industrial piece of equipment, and as mentioned, organic volatile solvents are not recommended in food processing [22]. With the aim of identifying other methodologies, a combination of mechanical treatments with different solid:solvent ratios was used in this study to break the cell wall and release lycopene.

All combinations of mechanical treatments used in this study reduced the size of FPG tissue in a different way, as each of the techniques is associated with different operating principles and mechanical forces. Differences in particle sizes after each mechanical treatment, associated with the forces exerted during each process can be observed in Figure 2 and Table 1. As reported by other authors, increasing the intensity of mechanical treatment produces the fragmentation of vegetable tissue, decreases the particle sizes of vegetable tissue and produces more homogeneous suspension [17,18,19,24].

A WB reduces particle size by the effect of the blade, which cuts the guava fruit (see Figure 2a) [20]. As observed in Figure 2a,d,g, larger tissue fragments and aggregates were still present when a WB was used to process FPG. The combination of WB + HSM destroyed the cellular structures. However, aggregates of cellular fragments are still present, as observed in Figure 2b,e,h. In HSM, a high rotor tip speed is responsible for the generation of shear, cavitation and turbulence stresses, leading to the breakdown of the plant material (see Figure 2b) [29], and, consequently, could be related to a more efficient size reduction than with a WB.

Finally, the sample processed by WB + UFFG still had aggregates of the cellular fragments, but the number of visible aggregates was significantly reduced in the samples, as shown in Figure 2c,f,i. This result is associated with the operation principle of this technique: two non-porous ceramic discs are separated by an adjustable distance. One disc is stationary, and the other rotates between 750 and 3000 rpm. The distance between the stationary and rotatory discs will affect the severity of the mechanical process, with lower distances increasing the compression, rolling and shearing forces and promoting a better disruption of the plant tissue (see Figure 2c) [30,31].

Differences in the size distribution of vegetable tissue associated with a combination of different treatments were also reported by Mert B. et al. (2012) during the ketchup process. The authors reported differences between the microstructure of the valve-homogenized sample and microfluidized sample. Although these two samples were homogenized at similar pressure, a more pronounced disintegration of the cellular material was observed in the microfluidized sample [19].

Differences in the size distribution of vegetable tissue observed by optical images were complemented by size distribution parameters. Table 1 lists the size distribution parameters of vegetable tissue observed in optical images of suspensions. The treatment WB + UFFG resulted statistically different to WB and WB + HSM. No statistical differences were evidenced between WB and WB + HSM, which can be associated with the heterogeneity of the particle sizes observed for both treatments in all FPG:SFO ratios (see Figure 2a–f). This size distribution heterogeneity can also be observed in the SD reported for D_mean_. In addition, D_mean_ and diameters D_10_, D_50_ and D_90_ (particle size at 10%, 50% and 90% in a cumulative size distribution, respectively) evidenced that particle size reduction is influenced by the FPG:SFO ratio. As observed in Table 1, the WB + UFFG treatment achieved the lowest D_mean,_ D_10_, D_50_, D_90_ and span at a 1:20 ratio. The solid: solvent ratio plays an important role in the creation of new surfaces by mechanical treatments. While the amount of SFO can affect the formation of new surfaces, higher solid concentration can increase the mechanical forces between the solid particles and improve the homogenization process [32]. This effect of the solid–solvent ratio in WB was also reported by Uetani and Yano (2011) [32]; they associated it with the amount of solvent required to create new surfaces. Uetani and Yano (2011) used a WB to reduce the size distribution of cellulosic fibers in Japanese cedar (*Cryptomeria japonica*) pulp [32]. They reported that the particle size decreased when they increased the pulp concentration up to 0.7 wt%, whereas it decreased over 1.1 wt% [32]. According to the authors, despite the fact that the chances of the pulp fibril sheeting each other were higher with the higher pulp concentration, increasing the solids concentrations above 1.1 wt% is associated with less solvent than required to create new surfaces by mechanical treatment. [32]. The solvent requirement to create new surfaces is also observed in this research for WB + UFFG, the size particle decreases as solvent increases as observed in Table 1.

In summary, the combination of mechanical treatments and the FPG:SFO ratio affects the particle size. In WB + UFFG, particle size reduced as solvent concentration increased, but WB + HSM were not affected by the FPG:SFO ratio. Subsequently, a spectrophotometric method was used to quantify lycopene directly in SFO and understand the relation between particle size distribution and lycopene extraction. Several studies [12,33,34,35] have demonstrated that the rapid extraction of lycopene followed by spectrophotometric determination is an acceptable measurement technique. The wavelength used in spectrophotometric measurements is associated with the solvent and analyte because chemical interactions between them can modify the maximal absorption wavelength [12,36]. Interactions between SFO and lycopene produced a redshift phenomenon (Figure 3b), in comparison to measurements with hexane:ethanol:acetone (Figure 3a). The maximal absorption wavelength increased from 472 to 483 nm because SFO increased the number of π,π * transitions in the sample, reducing the energy gap and increasing the wavelength [37]. Therefore, the absorbance of the samples was measured at 483 nm. This wavelength was identified as the maximum wavelength for lycopene on SFO, as shown in Figure 3b.

In Table 2, lycopene concentration is reported in mg lycopene/g FPG to normalize concentrations. The mechanical process introduces differences in lycopene extraction. The comparison of all treatments for each FPG:SFO ratio evidences that WB + UFFG 1:20 allowed higher release of lycopene from FPG (18.22 ± 1.83 mg/g FPG (*p* < 0.05)) than WB 1:20 or WB + HSM 1:20. As observed in Figure 2, WB and WB + HSM are treatments that lead to a size reduction of tissue fragments with a higher presence of cell aggregates. Therefore, portions of vegetable tissue from FPG with lycopene inside remained intact after WB and HSM + WB, while in WB + UFFG, separated cell clusters were generated [38].

Different FPG:SFO ratios for each treatment showed an effect of lycopene extraction. As mentioned above, solid–solvent ratio can improve the effect of mechanical forces responsible for lycopene extraction from cell tissue. The highest lycopene isolation was reached in WB + UFFG 1:20 ratio also related to the lowest D_mean_ for each treatment (*p* < 0.05). It is expected that lycopene concentration increases when the size of the vegetable tissue decreases. A lower particle size could enhance the mass transfer from the damaged plant cells to the solvent [17,19,38]. Juric et al. (2021) [17] reported that the mechanical disruption effect of high-pressure homogenization improved the mass transfer of lycopene into the oil phase [17].

Figure 4 presents the extraction yields, calculated as reported in Section 3.3.3. Increasing the intensity of mechanical forces improves the extraction; WB was the technique that reached lower extraction yields, and WB + UFFG achieved higher extraction yields. Mechanical treatments in combination with SFO isolated between 29% and 66% of the ultrasound-assisted method. Similar results were observed by Rahimi and Mikani (2019) [12]; they isolated lycopene from tomato by ultrasound with SFO and calculated the extraction yield in comparison with a CSE. The highest yield reported in this work was 87.25% [12]. It should be remarked that the techniques used in this research do not allow the use of organic solvents such as the hexane:acetone:ethanol mixture. The mechanical forces associated with the extraction techniques mentioned above increase the temperature above ambient values, and the organic solvents used have low evaporation temperatures, close to ambient temperatures [23,24].

The extraction methodology proposed in this paper is in line with some recent publications that reported a reduction in the use of organic solvents using mechanical treatments for lycopene extraction from different sources. Table 3 compares results from the current research work with other mechanically assisted lycopene extraction from tomato. Interestingly, the data in Table 3 show that the concentration of lycopene observed in this work is comparable with the results obtained through other mechanical processes. It is important to remark that the methodology proposed in this work avoids completely the use of organic solvents. Moreover, the maximal lycopene concentration achieved in this work was significantly higher than that obtained by a high-pressure homogenizer (HPH) water assisted [18] and by an HPH oil–water emulsion assisted [17]. HPH and UFFG are scalable techniques that can be introduced in food processing to release lycopene and produce it as a raw material or to increase the concentration of lycopene in a food product [39]. UFFG presents an advantage over the HPH because it is possible to work at higher concentrations, and this technique allows a higher process flow, which means less processing time. Velásquez-Cock et al. (2016) compared the HPH and UFFG to produce fibrillation in cellulose from banana rachis; they reported concentrations of approximately 2 wt% for UFFG, while the technical specifications of the HPH did not allow concentration to exceed 0.5 wt% [24].

## 3. Materials and Methods

Fresh pink guavas (*Psidium guajava* L.) (FPGs) were purchased from a local market in Medellín, Antioquia (Colombia). The FPGs were composed of 86.83 wt% moisture [40], 0.945 wt% protein [40], 0.215 wt% fat [40], 0.660 wt% ash [40], 4.680 wt% crude fiber [40] and 8.565 wt% carbohydrates. To assure a higher lycopene concentration, FPGs were selected in their last color stage of the color chart [25,41] (see Figure 5). Before their use, FPGs were rinsed and disinfected by immersion in a quaternary ammonium solution (200 ppm) for 15 min and rinsed again with fresh water. The fruits were stored at −18 °C using aluminum foil bags to avoid sample degradation.

Commercial SFO was purchased from a local supermarket in Medellín, Antioquia (Colombia). The organic volatile solvents, including hexane, ethanol and acetone, were analytical grade and acquired from Merck (Darmstadt, Germany), and analytical grade lycopene was acquired from Sigma-Aldrich (St. Louis, MO, USA).

### 3.1. Characterization of FPG

#### 3.1.1. Quantification of Lycopene in FPG

For lycopene quantification, 1 g of FPG previously blended for 2 min at 3000 rpm (VMO100, Vitamix, Olmsted Falls, OH, USA) was added to a mixture of hexane: ethanol: acetone (2:1:1), under reduced light conditions (final volume was 20 mL). Then, it was extracted for 10 min in an ultrasound bath at a 90% power and 37 Hz frequency. The absorbance of the decanted upper phase of samples was monitored at 473 nm against a blank using a UV spectrophotometer (Thermo Scientific Evolution 300). The same solvent mixture of hexane: ethanol: acetone (2:1:1) was used for the calibration curve. Lineal range of concentration of the calibration curve [0.067–0.767] mg/L (R^2^ = 0.996) [22].

#### 3.1.2. Light Microscopy of FPG

For light microscopy, FPG was cut into halves, and the pericarp and pulp were used to characterize chromoplast in guava using bright field microscopy. An optical microscope (Ci-L, Nikon, Tokyo, Japan) coupled with a digital camera (Nikon) was used. Images were collected using NIS-Elements software and were taken using an objective lens magnification of 10× and 40×. Additionally, a photograph of one of the halves was taken using a digital camera OIS of 12 MP, F/1.8 and QuadLED flash (Apple Inc, Cupertino, CA, USA) [25,42].

### 3.2. Lycopene Extraction from FPG

#### Extraction with SFO

The extraction of lycopene was performed using SFO as the solvent [10,11,12,13]. Three different ratios of FPG:SFO were evaluated: 1:5, 1:10 and 1:20. An initial suspension was produced by blending each FPG:SFO ratio for 2 min at 3000 rpm using a waring blender (WB) (VMO100, Vitamix). Then, each suspension was individually processed by high-shear mixing (HSM) or ultrafine friction grinding (UFG). HSM was performed at 10,000 rpm for 2 min with an Ultra-Turrax (TS 50, IKA), using an impeller model S50N-G45M, and UFFG was performed using a Super Masscolloider (Masuko Sangyo) at 3000 rpm for 20 passes at 4 different levels of separation between the discs (5 passes on each level): 1.5, 1, 0.5 and 0. The operation variables for each process were selected based on experience of authors in using these techniques to extract other biomolecules [20,22,23]. Photographs of each piece of equipment with a scheme of its operation principle are shown in Figure 6. The flow diagram of the processes performed in each sample is presented in Figure 7. The product of every mechanical treatment is composed of two phases: an aqueous phase composed of FPG pulp and an oil phase composed of SFO. Finally, SFO phase was separated from FPG pulp by centrifugation at 6000 rpm for 15 min (Hermle Labortechnik).

### 3.3. Characterization of SFO Enriched with Lycopene

#### 3.3.1. Light Microscopy of SFO Enriched with Lycopene

Microstructures of the obtained suspensions after each mechanical treatment were observed using an optical microscope (Ci-L, Nikon, Tokyo, Japan) coupled with a digital camera (Nikon). The 1:5 and 1:10 ratios were diluted 15 and 10 times, respectively, with SFO and mixed. Two droplets were placed on a glass slide and covered using a coverslip. Four images of every sample were collected using the software NIS-Elements. A segmentation procedure using ImageJ was used to measure the longest distance between any two points along the particle boundary, also known as Feret’s diameter [43]. On average, 160 particles were measured for each sample. Results from segmentation of optical images were used to measure the size distribution parameters D_mean_, D_10_, D_50_ and D_90_ and span.

#### 3.3.2. Quantification of Lycopene in SFO

The final concentration of lycopene in SFO was developed using UV–vis spectroscopy (Spectrophotometer UV-Vis Evolution 300, Thermo Scientific). A methodology without organic volatile solvents was used; it was extracted and measured in SFO. A lycopene calibration curve for standard lycopene was developed in SFO. SFO was used as blank in all samples and calibration curve. Lineal range of concentration of the calibration curve [0.356–1.358] mg Lycopene/100 g SFO (R^2^ = 0.999). After centrifugation, each sample was diluted 4.6 times in SFO before the measurement [12].

#### 3.3.3. Determination of Lycopene Extraction Yield

The yield of extraction was deliberated through Equation (1) where A is lycopene content (mg/g) in FPG quantified by mechanical treatment and SFO extraction, and A_0_ is the lycopene content (mg/g) in FPG quantified by ultrasound solvent extraction.
(1)$$\mathrm{Extraction}\;\mathrm{yield}=\frac{A}{A_{0}}100$$

#### 3.3.4. Statistical Analysis

All assays were performed in triplicate. Results are given as the mean of the three measurements and the standard deviation (±). Statistical analysis was performed using Statgraphics Centurion18. All treatments were compared using one-way analysis of variance (ANOVA). Significance was determined at 95% of confidence. A *p*-value < 0.05 was considered significant.

## 4. Conclusions

Three different mechanical treatments were evaluated to extract lycopene from fresh pink guavas (*Psidium guajava* L.) using SFO as a solvent. Two variables were evaluated in the lycopene extraction process, a mechanical treatment and FPG:SFO ratio, with the aim of identifying the process conditions that increased the lycopene extraction. The mechanical treatment by means of WB + UFFG generated lower particle sizes given the complex forces characteristic of UFFG and related to a higher concentration of lycopene in oil. Additionally, in the WB + UFFG treatment, the FPG:SFO ratio influenced the resultant particle size and final concentration of lycopene extracted in SFO since this ratio affected the creation of new surfaces during the mechanical treatment, and it increased the mechanical stresses between the solid particles. The process that achieved the highest concentrations of lycopene from guava involved an FPG:SFO ratio of 1:20 and the WB + UFFG mechanical treatment, 18.225 ± 1.83 mg lycopene/g FPG. This lycopene concentration was significantly higher than that reported by other authors, using an HPH for tomato peel and several solvents such as water, SFO, ethyl lactate and acetone. The yield of this treatment in comparison to the conventional treatment was 66%. For future work, it would be interesting to evaluate which other compounds besides lycopene are extracted by this methodology and the chemical stability of the oil and compare the yield of this process with that of the traditional process.

## Figures and Tables

**Figure 1 molecules-27-01038-f001:**
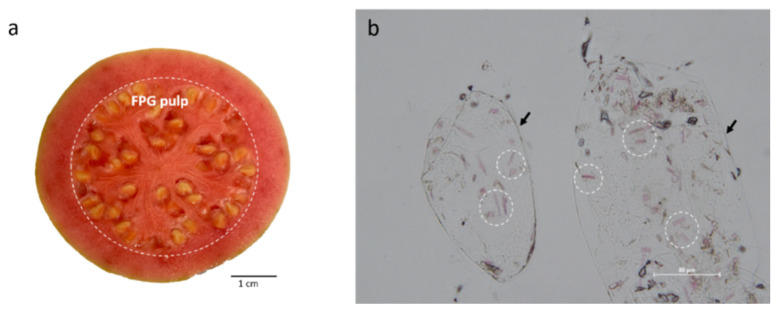
Guava (*Psidium guajava* L.) sections. (**a**) Photograph of the transversal section of a recently cut PFG. Pulp is enclosed in a dashed circle; (**b**) micrograph of FPG pulp 20×. Inside the dashed circles are lycopene crystals; the black arrows point to the plant cell wall.

**Figure 2 molecules-27-01038-f002:**
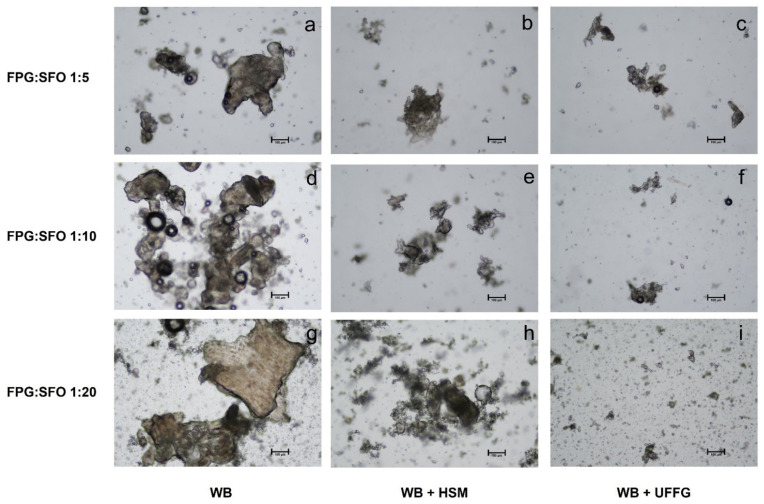
Suspensions obtained after the extraction processes: (**a**) WB 1:5, (**b**) WB + HSM 1:5, (**c**) WB + UFFG 1:5, (**d**) WB 1:10, (**e**) WB + HSM 1:10, (**f**) WB + HSM UFFG 1:10, (**g**) WB 1:20, (**h**) WB + UFFG 1:20 and (**i**) WB + UFFG 1:20 (scale size 100 μm, magnification 100×).

**Figure 3 molecules-27-01038-f003:**
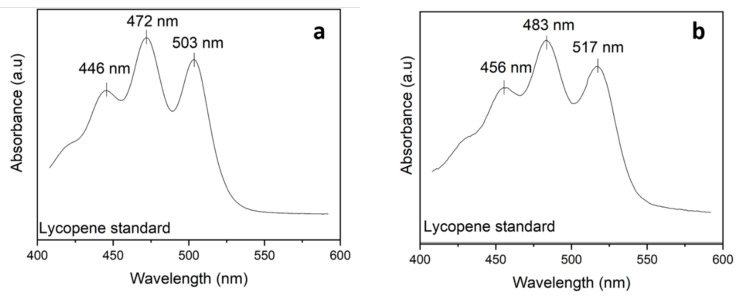
UV–vis spectra of lycopene standard in different solvents (1.36 mg lycopene/100 g solvent): (**a**) hexane:ethanol:acetone (2:1:1) and (**b**) SFO.

**Figure 4 molecules-27-01038-f004:**
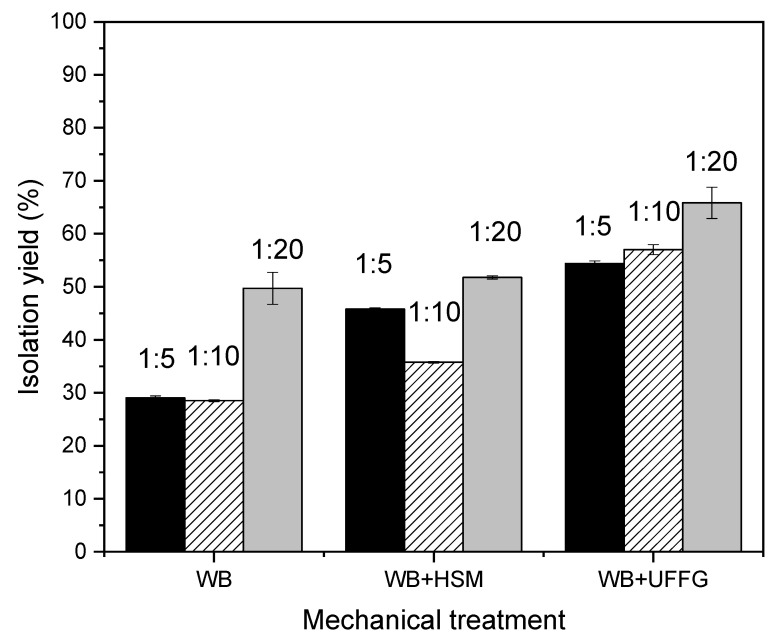
Lycopene isolation yield. Comparison between mechanical treatments and conventional treatments.

**Figure 5 molecules-27-01038-f005:**
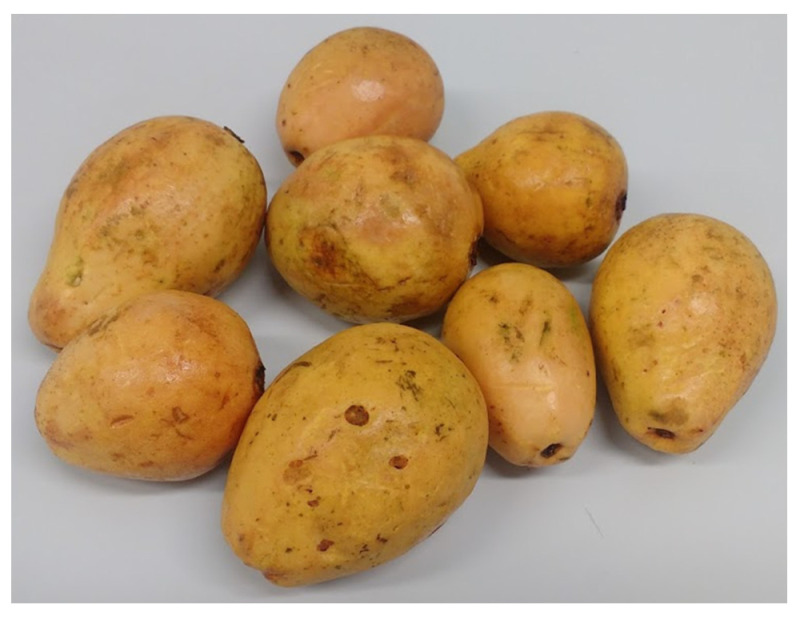
Photography of selected FPGs.

**Figure 6 molecules-27-01038-f006:**
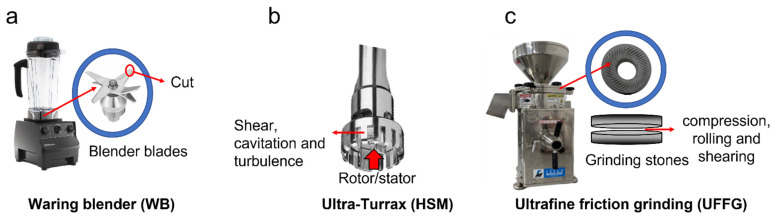
Photographs of the equipment used for mechanical treatment: (**a**) waring blender (WB), (**b**) ultrafine friction grinding (UFG) and (**c**) Ultra-Turrax (HSM).

**Figure 7 molecules-27-01038-f007:**
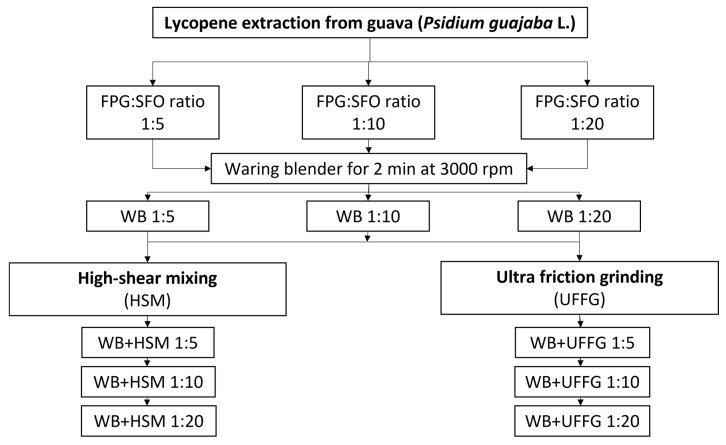
Scheme of the process assessed for lycopene extraction from FPG.

**Table 1 molecules-27-01038-t001:** Size distribution parameters of vegetable tissue from optical images of SFO enriched with lycopene.

**SFO:FPG Ratio**	**D_mean_ (μm)**	**D_10_ (μm)**	**D_50_ (μm)**	**D_90_ (μm)**
WB				
1:5	55.80 ± 1.96 ^A,a^	16.12 ± 1.43 ^A,a^	28.15 ± 0.76 ^A,a^	71.015 ± 6.49 ^A,a^
1:10	46.68 ± 16.51 ^A,a^	13.713 ± 0.72 ^A,a^	28,72 ± 4.95 ^A,a^	96.35 ± 23.53 ^A,a^
1:20	84.55 ± 35.03 ^A,a^	14.51 ± 1.08 ^A,a^	29.20 ± 5.15 ^A.a^	92.46 ± 7.69 ^A,a^
WB + HSM				
1:5	48.68 ± 2.68 ^A,a^	14.82 ± 0.84 ^A,a^	29.91 ± 6.16 ^A,a^	81.20 ± 13.40 ^A,a^
1:10	61.79 ± 0.83 ^A,b^	16.78 ± 4.46 ^A,a^	33.87 ± 7.24 ^A,a^	134.53 ± 44.12 ^A,a^
1:20	66.63 ± 1.75 ^A,b^	15.34 ± 0.19 ^A,a^	29.75 ± 0.78 ^A,a^	138.43 ± 15.12 ^B,a^
WB + UFFG				
1:5	78.35 ± 1.02 ^B,a^	17.03 ± 1.94 ^A,a^	34.24 ± 2.87 ^A,a^	191.65 ± 0.87 ^B,a^
1:10	41,85 ± 6.24 ^A,b^	14.48 ± 0.87 ^A,ab^	24.67 ± 2.10 ^A,b^	82.94 ± 7.28 ^A,b^
1:20	23.35 ± 0.65 ^B,c^	10.61 ± 0.13 ^B,b^	17.68 ± 1.15 ^B,c^	40.29 ± 6.54 ^C,c^

Different letters for each concentration indicate that the results are statistically different (*p* < 0.05). Capital letter—comparison at different treatments for each FPG:SFO ratio. Lowercase letter—comparison at different FPG:SFO for each treatment.

**Table 2 molecules-27-01038-t002:** Lycopene concentration after different mechanical treatments using several FPG:SFO ratios.

SFO:FPG Ratio	Mechanical Treatment		
	WB	WB + HSM	WB + UFFG
	(mg _lycopene_/g _FPG_)		
1:5	8.03 ± 0.23 ^A,a^	7.89 ± 0.11 ^B,a^	13.75 ± 1.88 ^C,a^
1:10	12.67 ± 0.12 ^A,b^	9.89 ± 0.10 ^B,b^	14.32 ± 0.19 ^C,b^
1:20	15.05 ± 0.30 ^A,b^	15.77 ± 0.59 ^A,c^	18.22 ± 1.8 ^B,c^

Different letters for each concentration indicate that the results are statistically different (*p* < 0.05). Capital letter—comparison at different treatments for each FPG:SFO ratio. All results are significantly different except WB at 1:10 and 1:20. Lowercase letter—comparison at different FPG:SFO for each treatment. All results are significantly different except WB and WB + HSM at 1:20.

**Table 3 molecules-27-01038-t003:** Comparison of the amount of lycopene recovered from FPG by WB + UFFG 1:20 (this work) with literature data, where different mechanical treatments and solvents were used.

Method Description	Lycopene Source	Extraction Solvent	Lycopene Concentration (mg/g)	Reference
WB + UFFG	FPG	SFO	18.21 ± 1.83 mg/g FPG	Current work
High-pressure homogenization water assisted	Fresh tomato residues	Water and ethyl lactate	4.00 mg/g tomato peel	[18]
High-pressure homogenization oil–water emulsion assisted	Fresh tomato	Water, SFO and acetone	3.94 ± 0.12 mg/g cream (oily phase)	[17]

## Data Availability

Not applicable.

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
