# Peer review of "An Edible Oil Enriched with Lycopene from Pink Guava (Psidium guajava L.) Using Different Mechanical Treatments"

_molecules, 2022, doi:10.3390/molecules27031038_

Round 1

Reviewer 1 Report

Please  check and correct sentences in row 101 and 242.

Author Response

Reviewer 1:

Comment

Answer

Please check and correct sentences in row 101 and 242.

Sentences in lines 101 and 242 were amended

Reviewer 2 Report

The authors describe the greener alternative for extraction of lycopene using sunflower oil (SFO) instead of traditionally employed solvents. The authors have also decided to stick to the methods based on emulsification and mixing using warring blender - WB, and followed by high shear mixing – HSM, and/or ultrafine friction grinding – UFGG) rather than ultrasound assisted extraction since the latter is not easily scalable in the industrial environment. They produced mixtures with various ratios of SFO and fresh pink guava – FPG (1:5, 1:10, and 1:20), and examined the impact of different mixing techniques (WB, WB+HSM, and WB+UFGG) and mixing ratios on microscopic properties of emulsions (presence of cellular structures and aggregates) as well as the yield of extracted lycopene in SFO.

The paper is correctly written. The literature is appropriately cited. The experiments were conducted in appropriate manner and described in a clear fashion. Conclusions have sound grounds based on the obtained results.

With few minor comments I consider the paper to meet the criteria for publishing in the Molecules.

Minor comments:

Line 65: Explain abbreviation HSM

Line 71: Please explain abbreviation HPH

Line 171: Plase change “shitting” into sheathing (or covering, protecting, sheeting). Shitting means defecating!

Table 2. Change “mg de lycopene” into English!

Change comma as a decimal separator into dot, through entire manuscript.

Author Response

Comment

Answer

Line 65: Explain abbreviation HSM

The abbreviation explanation was included in the text

Line 71: Please explain abbreviation HPH

The abbreviation explanation was included in the text

Line 171: Plase change “shitting” into sheathing (or covering, protecting, sheeting). Shitting means defecating!

The spelling mistake was amended

Table 2. Change “mg de lycopene” into English

The change suggested by the reviewer was made

Change comma as a decimal separator into dot, through entire manuscript.

Dot was used as decimal separator through the manuscript

Reviewer 3 Report

Line 37, add other solvents used in addition to hexane
Line 96. In addition to the form used (Charts) to identify the maturity stages of the guavas used, some other form was used, if so describe it
Line 111. Please enter the% of extraction by CFS, as a comparison with the data that this section describes
Line 174. Correct the word sufaces because I want to assume "surface"
Line 200. Use lycopene instead of Lycopene
Please improve figure 4, as the values are superimposed on the error bars
Line 300. Correct the word authors
Figure 6. Correct the figure caption according to what is described therein
Line 344. Correct themean
It is important to check if it is being referenced as indicated in the journal

Author Response

Comment

Answer

Line 37, add other solvents used in addition to hexane

Other solvents were added

Line 96. In addition to the form used (Charts) to identify the maturity stages of the guavas used, some other form was used, if so describe it

As mentioned in methodology only color char was used to identify the maturity stages of the guavas.

Line 111. Please enter the % of extraction by CFS, as a comparison with the data that this section describes

The concentration of lycopene by a conventional solvent extraction (CES) was included.

Line 174. Correct the word sufaces because I want to assume "surface"

The spelling mistake was amended

Line 200. Use lycopene instead of Lycopene

The change suggested by the reviewer was made

Please improve figure 4, as the values are superimposed on the error bars

Figure 4 was improved

Line 300. Correct the word authors

The spelling mistake was amended

Figure 6. Correct the figure caption according to what is described therein

The figure caption was modified according to the reviewer suggestion

Line 344. Correct themean

The spelling mistake was amended

It is important to check if it is being referenced as indicated in the journal

The reference style is according to the suggested by journal template